# The Association between Diabetes Mellitus, High Monocyte/Lymphocyte Ratio, and Survival in Endometrial Cancer: A Retrospective Cohort Study

**DOI:** 10.3390/diagnostics13010044

**Published:** 2022-12-23

**Authors:** Ruo-Shi Bing, Wing Lam Tsui, Dah-Ching Ding

**Affiliations:** 1Department of Medicine, Taipei Tzu Chi Hospital, Buddhist Tzu Chi Foundation, New Taipei City 23106, Taiwan; 2Department of Obstetrics and Gynecology, Hualien Tzu Chi Hospital, Buddhist Tzu Chi Foundation, Tzu Chi University, Hualien 970, Taiwan; 3Institute of Medical Sciences, College of Medicine, Tzu Chi University, Hualien 970, Taiwan

**Keywords:** endometrial cancer, survival, monocyte–lymphocyte ratio, lymphocyte, platelet

## Abstract

This retrospective cohort study aimed to evaluate the factors related to endometrial cancer (EC) prognosis and survival in eastern Taiwan. The study involved 48 patients diagnosed with EC who underwent hysterectomy-based surgery at Hualien Tzu Chi hospital between January 2011 and June 2021. The patients’ medical history and laboratory examination results were reviewed. Progression-free survival and overall survival were determined. Categorical variables were analyzed using the chi-square test, and continuous variables were analyzed using the independent *t*-test. The receiver operating characteristic curve was used to predict diagnostic value. Factors associated with cancer mortality were identified via Cox regression analysis (*p* < 0.05). Patients were divided into the death (*n* = 7) and survival (*n* = 41) groups. The median age of the patients was 56 years (range: 31–71 years). The median observation period was 33.29 months. Diabetes mellitus (DM) and monocyte/lymphocyte ratio (MLR) > 0.23886 were significantly associated with cancer mortality (*p* = 0.024 and *p* = 0.028, respectively). MLR-low and MLR-high groups exhibited 5-year overall survival rates of 96% and 60%, respectively, and 5-year progression-free survival rates of 96% and 41%, respectively. DM and MLR of >0.2386 were suggested to be associated with cancer death, poor overall survival, and progression-free survival.

## 1. Introduction

Endometrial cancer (EC) is the most common gynecological malignancy in developed countries, accounting for nearly 5% of the cancer cases [1]. In Taiwan, EC is ranked fifth in terms of female cancer incidence, with 2884 new cases in 2019. Elderly women with EC frequently present with postmenopausal vaginal bleeding. Therefore, the cancer is often diagnosed at an early stage [2]. Surgical treatments of EC include total hysterectomy with bilateral salpingo-oophorectomy and retroperitoneal lymphadenectomy [3]. Adjuvant therapy includes systemic chemotherapy and radiotherapy if advanced stages are noted [4].

Despite the early-stage detection of the disease, the 5-year overall survival (OS) of EC ranges from 74% to 91% [5]. However, the 5-year survival rate decreases to 57–66% or 20–26% for patients with stage III or IV of the disease, respectively, based on the 2009 International Federation of Gynecology and Obstetrics (FIGO) staging system [6]. Hence, it is important to identify the prognostic factors of EC to prevent premature death and improve overall patient outcome [7].

Numerous studies have identified the prognostic factors of EC, such as histological grading and type, myometrial invasion depth, lymphovascular invasion, lymph node status, age, serous and cervical involvement, tumor size, and stromal involvement [8]. Several biomarkers are also related to the prognosis of EC, such as TP53 mutations [9] and HF4 (human epididymis protein 4) [10]. A high monocyte/lymphocyte ratio (MLR) is associated with poor prognosis in several cancers, such as colorectal cancer [11], ovarian cancer [12], and lymphoma [13]. In the case of EC, studies have suggested that MLR is a solid independent prognostic predictive factor and could provide an additional value beyond serving as a traditional clinicopathological factor [14].

Apart from MLR, predisposing diabetes mellitus (DM), primarily type 2, may also affect the survival of patients with EC [15]. Many studies have suggested that the state of insulin resistance and subsequently, the state of hyperinsulinemia in patients increase the risk of endometrial carcinogenesis, ultimately leading to cancer development and lowering cancer survival [15]. Hence, our study aimed to evaluate the factors, including MLR and history of DM, related to endometrial cancer prognosis and survival in the eastern Taiwanese population.

## 2. Materials and Methods

### 2.1. Ethics

This study was approved by the Research Ethics Committee of Hualien Tzu Chi Hospital, Hualien, Taiwan (IRB 111-111-B). We retrospectively analyzed the data of all patients in our hospital who had EC (C54.1 of ICD-10-CM) and underwent hysterectomy-based surgical treatment between January 2011 and June 2021. The Research Ethics Committee of Hualien Tzu Chi Hospital waived the requirement for informed consent. This study was conducted in accordance with the Declaration of Helsinki. Relevant guidelines and regulations were implemented for all the methods.

### 2.2. Study Population

This retrospective study comprised 48 patients diagnosed with EC who underwent hysterectomy-based surgical treatment at our hospital between January 2011 and June 2021. Patients without full laboratory data, including complete blood count and differential count within two weeks before surgery, were excluded. Patients with active infections, autoimmune diseases, or hematological diseases were excluded from this study. EC diagnosis was confirmed by a pathologist.

### 2.3. Data Collection

In this study, we collected clinical information of the patients from the electronic medical records in the hospital information system. Patients who were diagnosed with malignant neoplasms of the endometrium (C54.1 ICD-10-CM) and who underwent hysterectomy-based surgery at our hospital were selected for the study. We collected information including (i) basic information such as age at surgery, body mass index (BMI), history of DM, and hypertension, as well as family history of gynecological cancer; (ii) pathological results including histological subtype, tumor grade, lymphovascular invasion (LVSI), lymph node (LN) invasion, clinical staging, FIGO staging, and immunohistochemistry stain of estrogen receptor (ER) and progesterone receptor (PR); (iii) laboratory data including complete blood count, absolute neutrophil count, absolute lymphocyte count, and absolute monocyte count; (iv) operation history, including surgery date, type, and route. Clinical and tumor grades were determined using the FIGO staging system. OS was defined as the time from the date of hysterectomy-based surgical treatment to the date of death or last follow-up. Progression-free survival (PFS) was defined as the time from the date of hysterectomy-based surgical treatment to the date of the first recurrence or last follow-up.

### 2.4. MLR, NLR and PLR Calculation

MLR was calculated by monocyte count divided by lymphocyte count. Neutrophil/lymphocyte ratio (NLR) was calculated by neutrophil count divided by lymphocyte count. Platelet/lymphocyte ratio (PLR) was calculated by platelet count divided by lymphocyte count.

### 2.5. Statistical Analysis

Categorical variables were analyzed using the chi-square test, and continuous variables were analyzed using the independent *t*-test. The receiver operating characteristic (ROC) curve was used to determine the largest area under the curve (AUC) to predict diagnostic value. Cox regression analysis was performed to investigate the factors associated with cancer mortality. All analyses were performed using SPSS 24.0 (IBM Corp., Armonk, NY, USA). Statistical significance was set at *p* < 0.05.

## 3. Results

### 3.1. Demographics

A total of 48 patients were ultimately eligible for the study and were divided into death (*n* = 7) and survival (*n* = 41) groups. The flowchart describing the patient selection process in the study is shown in Figure 1. All patients underwent hysterectomy-based surgery including bilateral salpingo-oophorectomy. The demographic characteristics of the patients are summarized in Table 1. The median age of the patients was 56 years (range: 31–71 years). The median observation period was 33.29 months. Demographic data including age, BMI, hypertension, stage, histology, tumor grade, ER, PR, LVSI, LN invasion, neutrophil/lymphocyte ratio (NLR, *p* = 0.085), and platelet/lymphocyte ratio (PLR, *p* = 0.725) were not significantly different between the two groups. However, the incidences of DM (*p* = 0.033) and MLR (*p* = 0.018) were significantly different between the two groups (Table 1). The median NLR, MLR, and PLR were 3.36 (range: 1–18), 0.23 (range: 0.1–0.9), and 176.67 (range: 59.21–697.25), respectively.

### 3.2. Cox Regression Analysis of the Factors Associated with Mortality

Table 2 lists the factors associated with EC-related mortality. DM and MLR > 0.23886 were significantly associated with cancer mortality (*p* = 0.024 and *p* = 0.028, respectively). In contrast, there was no significant association between cancer mortality and age, BMI, NLR, PLR, histology subtype, endometrial grade, ER, PR, LVSI, or lymph node invasion. Factors with a *p*-value < 0.1 were selected for adjustment in model 1. The analysis revealed that DM was associated with cancer mortality (*p* = 0.030). Due to the small number of patients with late FIGO stage in Adjusted Model 1, we further adjusted the factors in model 2. After adjustment in Model 2, an M/L ratio >0.2386 was associated with cancer mortality (*p* = 0.046).

### 3.3. ROC Curves of Various Ratios

ROC curves for the optimal cutoff values of NLR, MLR, and PLR are shown in Figure 2, Figure 3 and Figure 4. The cutoff value of NLR was 3.01 (AUC = 0.871). As for MLR and PLR, the cutoff value was 0.234 (AUC = 0.829) and 154.33 (AUC = 0.617), respectively.

### 3.4. Comparative Characteristics of High and Low NLR/MLR/PLR Groups

We further divided the patients into low and high NLR, MLR, and PLR groups based on the cutoff values (Table 3). The NLR was significantly associated with the incidence of DM (*p* = 0.008) and FIGO stage (*p* = 0.037). However, MLR was significantly associated with the FIGO stage *(p* = 0.002), histological subtype (*p* = 0.018), lymphovascular invasion (*p* = 0.015), and lymph node invasion (*p* = 0.035). There was no significant association between the PLR and clinicopathological factors.

### 3.5. OS and PFS

OS and PFS were evaluated using the Kaplan–Meier method, stratified by the MLR cutoff value (Figure 5 and Figure 6). A higher MLR was associated with a significantly worse OS (*p* = 0.006) and PFS (*p* = 0.004) than a lower MLR. The 5-year OS rates of MLR-low and MLR-high groups were 96% and 60%, respectively. The 5-year PFS rates of MLR-low and MLR-high groups were 96% and 41%, respectively.

## 4. Discussion

In this retrospective cohort study exploring the effects of MLR and DM on the survival of patients with EC, we found that the MLR was significantly different between the two groups of patients in terms of EC survival. The statistical analysis showed that an MLR of >0.2386 was associated with cancer-related death, worse OS, and worse PFS. DM was also associated with cancer-related deaths. In contrast, other risk factors such as age, BMI, stage, tumor grade, LVSI, and lymph node invasion were not associated with cancer-related death in this study. The MLR represents a delicate balance between the two blood cells (monocytes and lymphocytes) and is a novel indicator of inflammation [16]. Numerous studies have suggested that MLR is an independent and crucial prognostic factor for various diseases [17]. It provides valuable information regarding patients’ abnormal immune status against diseases [18].

In general, lymphocyte count is a useful indicator of host immunity [19]. It is crucial to the immune surveillance and defense system against cancer cells. The tumor microenvironment not only consists of cancerous cells but also all other types of inflammatory cells. CD8+ T cells eliminate tumor cells through the action of granzymes and perforins [20]. CD4+ T cells induce and regulate T cell-mediated cytotoxicity [21]. On the contrary, they also release various cytokines, such as INF-*γ*, after being activated by tumor antigens, preventing macrophages from migrating away from the tumor antigens, as well as TNF-ɑ, IL-4, and IL-5. Therefore, a high lymphocyte count is favorable for patients with cancer [22].

However, monocytes are released by the bone marrow and circulate in the blood [23]. Following recruitment to tumor tissues, they transform into macrophages, specifically M1 and more importantly, M2 macrophages, which are also known as tumor-associated macrophages [23]. M1 macrophages are known to exhibit antitumor activity. However, tumor-associated macrophages produce vast amounts of anti-inflammatory cytokines, angiogenic factors, and metalloproteases, which aid the growth of tumor cells [24]. Therefore, a high monocyte count is unfavorable for patients and imparts a negative effect on patient prognosis [22].

The results of our study were consistent with those of previous studies. Cong et al. found that a high MLR (>0.22) negatively affected the survival of endometrial cancer patients [25]. Song et al. showed similar results, in which the cutoff point of MLR was 0.19 and an MLR higher than that was associated with cancer recurrence and mortality [26]. Another study by Holub et al. reported an MLR cutoff of 0.18, with higher values leading to a poorer prognosis and a decrease in OS [27]. Leng et al. conducted a meta-analysis that included more than 5000 endometrial cancer patients and found that NLR or PLR was associated with OS and disease-free survival (DFS) and that NLR was associated with PFS only in univariate analysis but also that MLR was not associated with OS or DFS [28]. However, there were only two references included in the above study regarding MLR, and it found no correlation between MLR and OS or DFS. We thought the number of studies included in the meta-analysis was too small to drive a conclusion. Therefore, the current study could add to the current knowledge about the association between MLR and OS or DFS. There are still uncertainties regarding the cutoff point for MLR and its impact on the survival of EC, as each study had unique qualities in its study group. Nonetheless, there is no doubt that most recent studies unanimously point to the fact that a high MLR is an independent poor prognostic factor for EC.

A previous study by Luo et al. reported different results regarding diabetes and survival of patients with EC. In the study, although initial results showed that self-reported diabetes was negatively correlated to OS in EC, the significant relationship between diabetes and EC survival was no longer present after adjusting for BMI, treatment, medication, and duration of disease [29]. This indicates that the relationship between diabetes and EC involves various confounding factors that were not analyzed in our study, leading to the results being different from ours. Metformin was reported to increase survival in women with EC [30]. In our study, all DM patients had type 2 DM and received pharmacotherapy. However, the oral hypoglycemic agents varied among the patients.

EC stage and grade are known to be related to prognosis and survival [14]. In fact, another study using the SEER database confirmed that the EC stage and grade were correlated with survival [31]. However, the correlation between EC stage, grade, and survival in our study was only a trend but did not reach a statistical significance. We speculate that this might be due to our study’s small number of deaths, particularly when cancer itself may not be the cause.

This retrospective cohort study collected information from a hospital database. The patients included in our study had detailed medical records and provided their complete profiles, characteristics, and laboratory data. Therefore, recall bias was not observed. This also provided us with the opportunity to explore multiple prognostic factors simultaneously. However, there are several limitations in this study that need to be addressed, specifically its relatively small sample size. It is also a retrospective cohort study, hence confounding bias may affect the final result. Nevertheless, the results of this study have shed light on the importance of MLR in evaluating patient prognosis and mortality in EC. Large-scale studies should be conducted to further evaluate the accuracy and importance of MLR as well as its role in predicting the prognosis of patients with EC.

## 5. Conclusions

We demonstrated that an MLR of more than 0.2386 was suggested to be associated with cancer death, worse OS, and worse PFS compared to MLR under 0.2386. DM was also found to be associated with cancer-related deaths. Further large-scale trials are needed to corroborate the findings in this study.

## Figures and Tables

**Figure 1 diagnostics-13-00044-f001:**
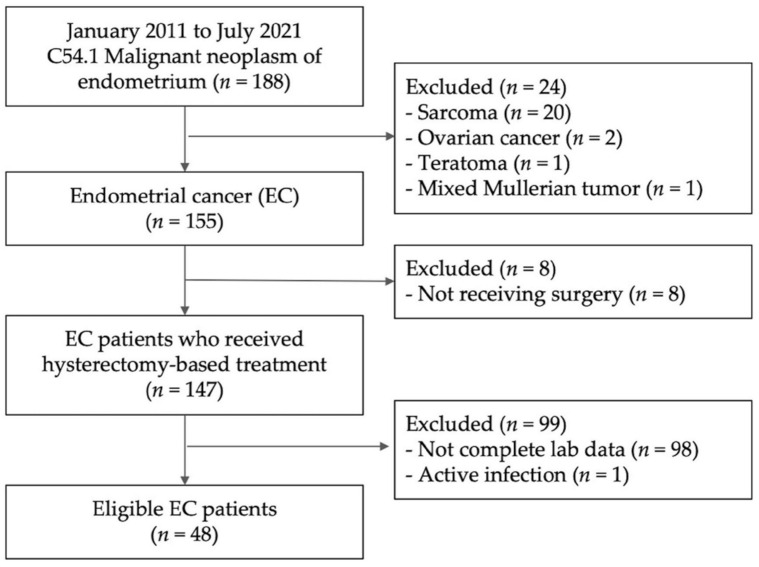
Flowchart describing the patient selection for the study. EC: endometrial cancer.

**Figure 2 diagnostics-13-00044-f002:**
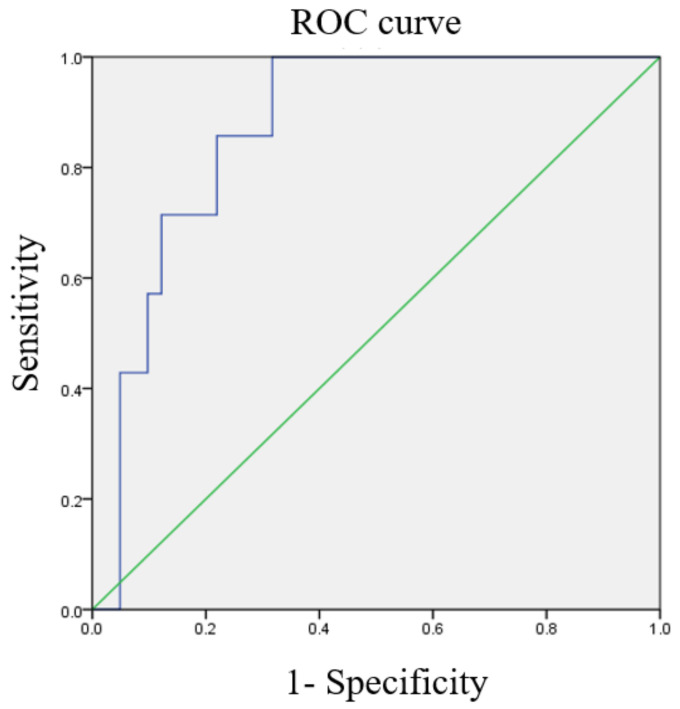
The receiver operating characteristic (ROC) curve of the neutrophil/lymphocyte ratio predicts cancer death (area under the curve [AUC] = 0.871).

**Figure 3 diagnostics-13-00044-f003:**
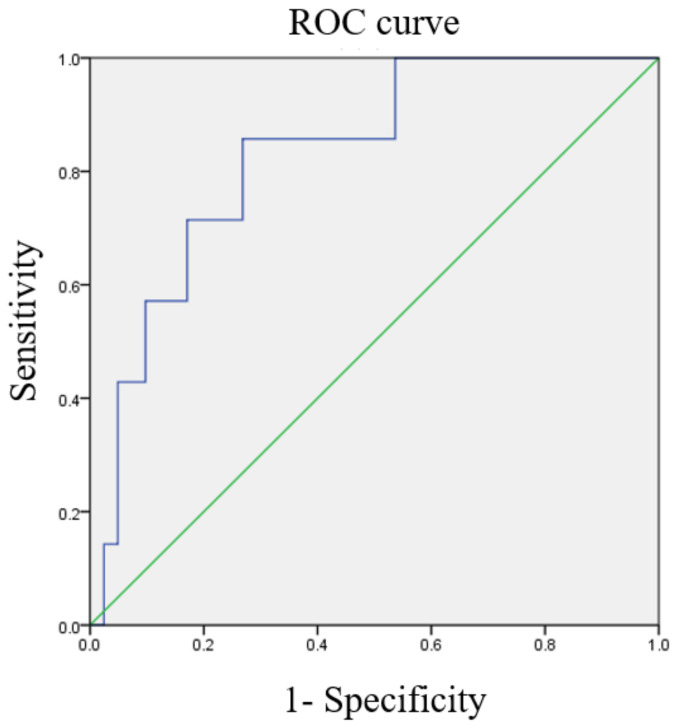
The receiver operating characteristic (ROC) curve of the monocyte/lymphocyte ratio predicts cancer death (area under the curve [AUC] = 0.829).

**Figure 4 diagnostics-13-00044-f004:**
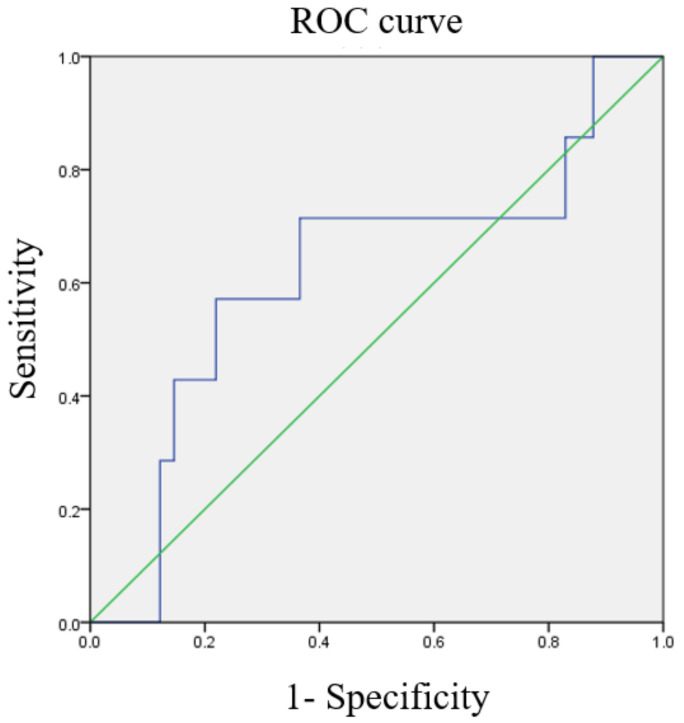
The receiver operating characteristic (ROC) curve of the platelet/lymphocyte ratio predicts cancer death (area under the curve [AUC] = 0.617).

**Figure 5 diagnostics-13-00044-f005:**
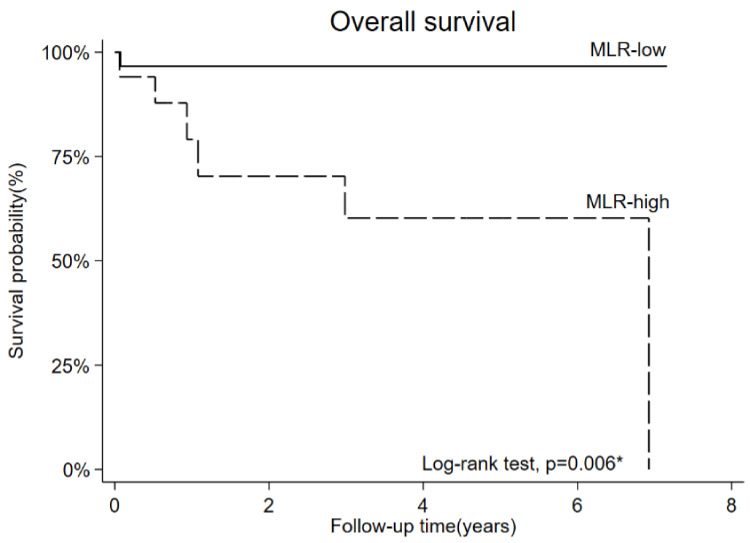
Overall survival stratified by high and low monocyte/lymphocyte ratio (MLR) (5-year survival rate = 96% and 60% in the MLR-low and MLR-high groups, respectively).

**Figure 6 diagnostics-13-00044-f006:**
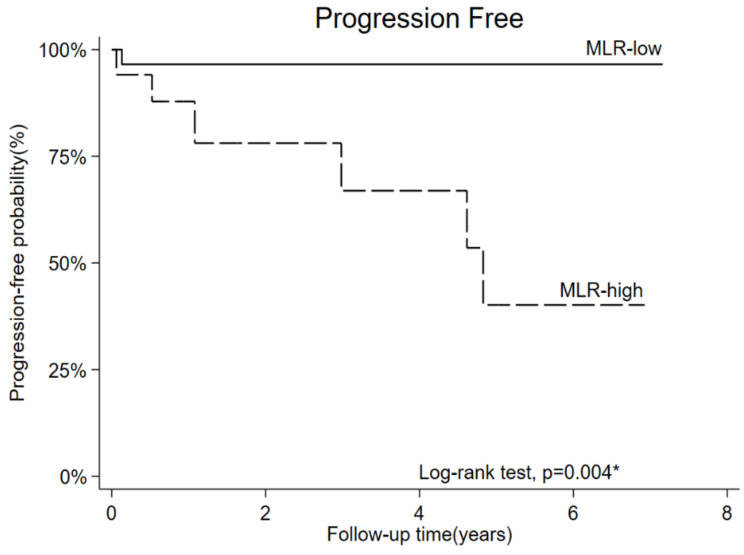
Progression-free survival stratified by high and low monocyte/lymphocyte ratio (MLR) (5-year progress-free survival rate = 96% and 41% in the MLR-low and MLR-high groups, respectively).

**Table 1 diagnostics-13-00044-t001:** Demographic characteristics of patient data involved in the study (*n* = 48).

	Death		Total	*p*-Value
	No	Yes		
N	41	7	48	
Age	55.95 ± 8.83	61.57 ± 7.32	56.77 ± 8.79	0.119
BMI	28.38 ± 5.13	30.97 ± 4.30	28.76 ± 5.06	0.215
DM (%)	11 (26.8%)	5 (71.4%)	16 (33.3%)	0.033 *
Hypertension (%)	23 (56.1%)	4 (57.1%)	27 (56.3%)	1.000
Stage (%)	-	-	-	0.288
1	28 (68.3%)	3 (42.9%)	31 (64.6%)	
2	5 (12.2%)	1 (14.3%)	6 (12.5%)	
3	5 (12.2%)	2 (28.6%)	7 (14.6%)	
4	3 (7.3%)	1 (14.3%)	4 (8.3%)	
Histology subtype (%)	-	-	-	0.684
Mixed cell carcinoma	4 (9.8%)	1 (14.3%)	5 (10.4%)	
Endometrioid	32 (78.0%)	5 (71.4%)	37 (77.0%)	
Serous carcinoma	4 (9.8%)	1 (14.3%)	5 (10.4%)	
Clear cell carcinoma	1 (2.4%)	0 (0.0%)	1 (2.1%)	
Tumor grade (%) N = 48	-	-	-	0.875
1	21 (51.2%)	4 (57.1%)	25 (52.1%)	
2	9 (22.0%)	1 (14.3%)	10 (20.8%)	
3	11 (26.8%)	2 (28.6%)	13 (27.1%)	
Immunohistochemistry				
ER				0.363
0	6 (14.6%)	2 (28.6%)	8 (16.7%)	
1+	11 (26.8%)	3 (42.9%)	14 (29.2%)	
2+	11 (26.8%)	0 (0.0%)	11 (22.9%)	
3+	13 (31.7%)	2 (28.6%)	15 (31.1%)	
PR N = 46				0.681
0	5 (13.8%)	1 (14.3%)	6 (13.0%)	
1+	8 (20.5%)	3 (42.9%)	11 (23.9%)	
2+	10 (25.6%)	1 (14.3%)	11 (23.9%)	
3+	14 (41.0%)	2 (28.6%)	18 (39.1%)	
Lymphovascular invasion (%) N = 47	19 (47.5%)	5 (71.4%)	24 (51.1%)	0.416
LN invasion (%) N = 41	6 (17.1%)	2 (33.3%)	8 (19.5%)	0.578
NLR	3.04 ± 3.10	5.23 ± 2.68	3.36 ± 3.11	0.085
MLR	0.21 ± 0.15	0.36 ± 0.15	0.23 ± 0.16	0.018 *
PLR	174.14 ± 123.41	191.47 ± 92.19	176.67 ± 118.68	0.725

Data are presented as *n* or mean ± standard deviation; * *p*-value < 0.05 was considered statistically significant after the test; NLR: neutrophil/lymphocyte ratio, MLR: monocyte/lymphocyte ratio, PLR: platelet/lymphocyte ratio, LN: lymph node, BMI: body mass index, DM: diabetes mellitus, ER: estrogen receptor, PR: progesterone receptor. 0: no staining, 1+: weak staining, 2+: intermediate staining, 3+: strong staining.

**Table 2 diagnostics-13-00044-t002:** Factors associated with mortality (*n* = 48).

	Crude		Adjusted (Model 1)		Adjusted (Model 2)	
	HR (95% CI)	*p*-Value	HR (95% CI)	*p*-Value	HR (95% CI)	*p*-Value
Age	1.12 (0.99, 1.27)	0.073	1.02 (0.88, 1.19)	0.780	1.04 (0.90, 1.20)	0.636
BMI	1.12 (0.96, 1.30)	0.145				
DM (Yes vs. No)	11.89 (1.39, 102.02)	0.024 *	21.54 (1.34, 344.93)	0.030 *	8.27 (0.81, 84.25)	0.074
NLR > 3.0995	274.63 (0.06, 1,236,601.52)	0.191				
MLR > 0.2386	10.87 (1.30, 90.83)	0.028 *	16.05 (0.9, 286.66)	0.059	8.88 (1.03, 76.28)	0.046 *
PLR > 154.3309	3.34 (0.65, 17.24)	0.150				
Stage	-	-	-	-		
1	Reference	NA	Reference	NA		
2	0.99 (0.09, 10.88)	0.993	19.78 (0.66, 590.95)	0.085		
3	4.72 (0.75, 29.76)	0.098	2.16 (0.24, 19.29)	0.489		
4	3.66 (0.36, 36.82)	0.270	3.19 (0.25, 41.03)	0.373		
Histology	-	-				
Mixed cell carcinoma	Reference	NA				
Endometrioid	0.43 (0.05, 3.86)	0.451				
Serous carcinoma	0.85 (0.05, 13.68)	0.910				
Clear cell carcinoma	0.00 (NA)	0.991				
Tumor grade	-	-				
1	Reference	NA				
2	1.05 (0.19, 5.89)	0.957				
3	0.56 (0.06, 5.08)	0.605				
ER	-	-				
0	References	NA				
1+	0.43 (0.05, 3.86)	0.451				
2+	0.85 (0.05, 13.86)	0.910				
3+	0.85 (0.05, 13.68)	0.991				
PR N = 46	-	-				
0	References	NA				
1+	1.33 (0.14, 12.99)	0.808				
2+	0.41 (0.03, 6.61)	0.520				
3+	0.27 (0.02, 4.43)	0.361				
Lymphovascular invasion (Yes vs. No)	1.52 (0.28, 8.43)	0.630				
LN invasion (Yes vs. No)	3.23 (0.53, 19.52)	0.202				

Data are presented as hazard ratio (HR; 95% CI); * *p*-value < 0.05 was considered statistically significant after the test. Crude: age, DM, BMI, N/L, M/L, P/L, stage, histology, grade, lymphovascular invasion, LN invasion; Adjusted Model 1: age, DM, M/L > 0.2386, stage; Adjusted Model 2: age, DM, M/L > 0.2386; DM: diabetes mellitus, BMI: body mass index, NLR: neutrophil/lymphocyte ratio, MLR: monocyte/lymphocyte ratio, PLR: platelet/lymphocyte ratio, LN: lymph node, ER: estrogen receptor, PR: progesterone receptor. 0: no staining, 1+: weak staining, 2+: intermediate staining, 3+: strong staining.

**Table 3 diagnostics-13-00044-t003:** Clinical and pathological characteristics of patients with EC in high and low NLR/MLR/PLR groups (*n* = 48).

Characteristic	NLR-Low	NLR-High	*p*-Value	MLR-Low	MLR-High	*p*-Value	PLR-Low	PLR-High	*p*-Value
N	33	15		31	17		28	20	
Age	55.39 ± 9.39	59.80 ± 6.59	0.108	55.61 ± 9.75	58.88 ± 6.41	0.221	55.57 ± 9.47	58.45 ± 7.65	0.268
BMI	28.61 ± 5.38	29.09 ± 4.40	0.767	28.16 ± 5.03	29.85 ± 5.08	0.271	29.14 ± 4.87	28.23 ± 5.38	0.543
DM (%)	7 (21.2%)	9 (60.0%)	0.008 *	9 (29.0%)	7 (41.2%)	0.393	8 (28.6%)	8 (40.0%)	0.408
Stage (%)			0.037 *			0.002 *			0.244
1	25 (75.8%)	6 (40.0%)		24 (77.4%)	7 (41.2%)		21 (75.0%)	10 (50.0%)	
2	4 (12.1%)	2 (13.3%)		5 (16.1%)	1 (5.9%)		3 (10.7%)	3 (15.0%)	
3	3 (9.1%)	4 (26.7%)		2 (6.5%)	5 (29.4%)		2 (7.1%)	5 (25.0%)	
4	1 (3.0%)	3 (20.0%)		0 (0.0%)	4 (23.5%)		2 (7.1%)	2 (10.0%)	
Histology (%)			0.096			0.018 *			0.175
Mixed cell carcinoma	3 (9.1%)	2 (13.3%)		2 (6.5%)	3 (17.6%)		3 (10.7%)	2 (10.0%)	
Endometrioid	28 (84.8%)	9 (60.0%)		28 (90.3%)	9 (52.9%)		24 (85.7%)	13 (65.0%)	
Serious carcinoma	1 (3.0%)	4 (26.7%)		1 (3.2%)	4 (23.5%)		1 (3.6%)	4 (20.0%)	
Clear cell carcinoma	1 (3.0%)	0 (0.0%)		0 (0.0%)	1 (5.9%)		0 (0.0%)	1 (5.0%)	
ER			1.000			0.974			0.271
0	5 (15.2%)	3 (20.0%)		5 (16.1%)	3 (17.6%)		3 (10.7%)	5 (25.0%)	
1+	10 (30.3%)	4 (26.7%)		9 (29.0%)	5 (29.4%)		8 (28.6%)	6 (20.0%)	
2+	8 (24.2%)	3 (20.0%)		8 (25.8%)	3 (17.6%)		9 (32.1%)	2 (10.0%)	
3+	10 (30.0%)	5 (33.3%)		9 (29.0%)	6 (35.3%)		8 (28.6%)	7 (35.0%)	
PR			0.536			0.334			0.951
0	3 (9.7%)	3 (20.0%)		2 (6.9%)	4 (23.5%)		3 (11.5%)	3 (15.0%)	
1+	7 (22.6%)	4 (26.7%)		6 (20.7%)	5 (29.4%)		6 (23.1%)	5 (25.0%)	
2+	7 (22.6%)	4 (26.7%)		8 (27.6%)	3 (17.6%)		7 (26.9%)	4 (20.0%)	
3+	14 (45.2%)	4 (26.7%)		13 (44.8%)	5 (29.4%)		10 (38.5%)	8 (40.0%)	
Lymphovascular invasion (%) *n* = 47	14 (43.8%)	10 (66.7%)	0.143	11 (36.7%)	13 (76.5%)	0.015 *	15 (53.6%)	9 (47.4%)	0.676
LN invasion (%) *n* = 41	3 (11.1%)	5 (35.7%)	0.097	2 (7.7%)	6 (40.0%)	0.035 *	4 (16.7%)	4 (23.5%)	0.698

Data are presented as *n* or mean ± standard deviation. * *p*-value < 0.05 was considered statistically significant after the test. DM: diabetes mellitus, BMI: body mass index, NLR: neutrophil/lymphocyte ratio, MLR: monocyte/lymphocyte ratio, PLR: platelet/lymphocyte ratio, LN: lymph node.

## Data Availability

Not applicable.

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
