# Peer review of "The Association between Diabetes Mellitus, High Monocyte/Lymphocyte Ratio, and Survival in Endometrial Cancer: A Retrospective Cohort Study"

_diagnostics, 2022, doi:10.3390/diagnostics13010044_

Round 1

Reviewer 1 Report

The manuscript "The association between diabetes mellitus, high monocyte/lymphocyte ratio, and survival in endometrial cancer: a retrospective cohort study" is interesting and well-written. The methodology is adequate and well-structured, and the results are presented very clearly. A final review for possible small mistakes is advisable such as the period found before the "Ethics" subtitle. 

It is also advisable to rephrase the conclusion in order to be a bit more conservative, as you state, large-scale trials are needed to corroborate the findings of the study and, as it has a small sample size, the study is limited (this is correctly stated in the discussion). The conclusion that the study demonstrates that an MLR of more than 0.2386 was associated with cancer death, worse OS, and worse PFS compared to MLR under 0.2386 is too bold, maybe say that the study suggests this association. 

Author Response

Reviewer 1

The manuscript "The association between diabetes mellitus, high monocyte/lymphocyte ratio, and survival in endometrial cancer: a retrospective cohort study" is interesting and well-written. The methodology is adequate and well-structured, and the results are presented very clearly. A final review for possible small mistakes is advisable such as the period found before the "Ethics" subtitle. 

Response: We deleted the period before the “Ethics” subtitle. 

It is also advisable to rephrase the conclusion in order to be a bit more conservative, as you state, large-scale trials are needed to corroborate the findings of the study and, as it has a small sample size, the study is limited (this is correctly stated in the discussion). The conclusion that the study demonstrates that an MLR of more than 0.2386 was associated with cancer death, worse OS, and worse PFS compared to MLR under 0.2386 is too bold, maybe say that the study suggests this association.

Response: We revised the conclusion accordingly. 

Reviewer 2 Report

In the present study based on a small cohort of EC patients, authors demonstrated that MLR and DM are related to a worse prognosis. Observed results are potentially interesting however, the following issues should be more clearly explained:

-          Materials and methods: authors should define more in deep how MLR is calculated

-          Results: it is know well established that EC prognosis is strictly related to the TCGA subgroups of endometrial cancer (POLE – MMR – p53…) therefore authors should also include informations on the molecular classes of the studied cohort or at least include immunohistochemical results for p53 and MMR proteins.

-          Table 1-2: endometrioid with squamous differentiation should be incorporate in the endometrioid group since they are the same entity.

Author Response

Reviewer 2. 

In the present study based on a small cohort of EC patients, authors demonstrated that MLR and DM are related to a worse prognosis. Observed results are potentially interesting however, the following issues should be more clearly explained:

-          Materials and methods: authors should define more in deep how MLR is calculated

Response: We added how to calculate MLR in the Method section. 

-          Results: it is know well established that EC prognosis is strictly related to the TCGA subgroups of endometrial cancer (POLE – MMR – p53…) therefore authors should also include informations on the molecular classes of the studied cohort or at least include immunohistochemical results for p53 and MMR proteins.

Response: We added ER and PR immunohistochemical results in the result section. Due to the small sample size, we did not include p53 and MMR protein in this study.

-          Table 1-2: endometrioid with squamous differentiation should be incorporate in the endometrioid group since they are the same entity.

Response: We combined these two categories into one.